# How effective is extracorporeal life support for patients with out-of-hospital cardiac arrest initiated at the emergency department? A systematic review and meta-analysis

Wachira Wongtanasarasin[1,2]*, Sarunsorn Krintratun[1], Witina Techasatian[3], Daniel K. Nishijima[2]

1 Department of Emergency Medicine, Faculty of Medicine, Chiang Mai University, Chiang Mai, Thailand, 2 Department of Emergency Medicine, University of California Davis School of Medicine, Sacramento, CA, United States of America, 3 Department of Medicine, John A. Burns School of Medicine, University of Hawai'i, Honolulu, HI, United States of America

* wachir_w@hotmail.com

**Data Availability Statement:** All relevant data are within the paper and its Supporting Information files.

## Abstract

### Background

Extracorporeal cardiopulmonary resuscitation (ECPR) is commonly initiated for adults experiencing cardiac arrest within the cardiac catheterization lab or the intensive care unit. However, the potential benefit of ECPR for these patients in the emergency department (ED) remains undocumented. This study aims to assess the benefit of ECPR initiated in the ED for patients with out-of-hospital cardiac arrest (OHCA).

### Methods

We conducted a systematic review and meta-analysis of studies comparing ECPR initiated in the ED versus conventional CPR. Relevant articles were identified by searching several databases including PubMed, EMBASE, Web of Science, and Cochrane collaborations up to July 31, 2022. Pooled estimates were calculated using the inverse variance heterogeneity method, while heterogeneity was evaluated using Q and $I^2$ statistics. The risk of bias in included studies was evaluated using validated bias assessment tools. The primary outcome was a favorable neurological outcome at hospital discharge, and the secondary outcome was survival to hospital discharge or 30-day survival. Sensitivity analyses were performed to explore the benefits of ED-initiated ECPR in studies utilizing propensity score (PPS) analysis. Publication bias was assessed using Doi plots and the Luis Furuya-Kanamori (LFK) index.

### Results

The meta-analysis included a total of eight studies comprising 51,173 patients. ED-initiated ECPR may not be associated with a significant increase in favorable neurological outcomes (odds ratio [OR] 1.43, 95% confidence interval [CI] 0.30–6.70, $I^2$ = 96%). However, this intervention may be linked to improved survival to hospital discharge (OR 3.34, 95% CI 2.23–

**Funding:** We confirm that this study was supported by the National Center for Advancing Translational Sciences, National Institutes of Health, through grant number UL1 TR001860. The funders had no role in study design, data collection and analysis, decision to publish, or preparation of the manuscript.

**Competing interests:** The authors have declared that no competing interests exist.

5.01, $I^2$ = 17%). Notably, when analyzing only PPS data, ED-initiated ECPR demonstrated efficacy for both favorable neurological outcomes (OR 1.89, 95% CI 1.26–2.83, $I^2$ = 21%) and survival to hospital discharge (OR 3.37, 95% CI 1.52–7.49, $I^2$ = 57%). Publication bias was detected for primary (LFK index 2.50) and secondary (LFK index 2.14) outcomes.

## Conclusion

The results of this study indicate that ED-initiated ECPR may not offer significant benefits in terms of favorable neurological outcomes for OHCA patients. However, it may be associated with increased survival to hospital discharge. Future studies should prioritize randomized trials with larger sample sizes and strive for homogeneity in patient populations to obtain more robust evidence in this area.

## Introduction

Approximately 350,000 individuals die from out-of-hospital cardiac arrest (OHCA) in the United States [1]. Despite significant advances in treatments such as rapid reperfusion, aggressive post-resuscitation care, and targeted temperature management (TTM), the rates of neurologically intact survival to hospital discharge remain alarmingly low, ranging from 6% to 9% [2–5].

Extracorporeal cardiopulmonary resuscitation (ECPR), involving the use of veno-arterial extracorporeal membrane oxygenation (ECMO) during ongoing cardiopulmonary resuscitation (CPR), has been employed for refractory OHCA for several decades [6–8]. However, the efficacy of EPCR has not been definitively established [9]. Prior studies have reported mixed findings, with some suggesting that ECPR may lead to improved mortality and neurological outcomes in OHCA patients compared to conventional CPR [10–12]. Nevertheless, other articles have demonstrated minimal or no effect on survival, while systematic reviews on this topic have produced conflicting results [10, 12–15]. These discrepancies are likely attributable to the included studies' inherent heterogeneity and selection method variations, resulting in survival rates ranging widely from 15% to 50% [10, 13, 16, 17].

Effective and safe ECPR necessitates specific medical skills and a well-trained team [18, 19]. Traditionally, EPCR is initiated in the cardiac catheterization lab or the intensive care unit, given the availability of specialized equipment and experienced physicians [20]. However, the emergency department (ED) typically serves as the initial point of care for patients undergoing CPR upon their arrival at the hospital [8]. Innovations in percutaneous vascular cannula insertion, centrifugal pump techniques, and extracorporeal devices have facilitated the application of ECPR in the ED under the guidance of trained emergency physicians [8, 11, 21–23]. Rapid stabilization in the ED assumes a pivotal role in sustaining critically ill patients and enables prompt decision-making for individuals with a limited prognosis [8, 11].

In this study, we aimed to conduct a systematic review and meta-analysis to evaluate the effectiveness of ED-initiated ECPR for OHCA patients. Our hypothesis posited that ED-initiated ECPR for patients with OHCA would be associated with improved survival and neurological outcomes.

## Methods

This systematic review adhered to the Preferred Reporting Items for Systematic Reviews and Meta-Analyses statements [24]. The study protocol was registered with the PROSPERO

website before the data collection (Registration ID: CRD42022343782, http://www.crd.york.ac.uk/PROSPERO).

## Search strategy and study selection

We systematically searched four databases, including PubMed, EMBASE, Web of Science, and Cochrane Collaboration, from their inception until July 31, 2022. Our search strategy did not impose any language restrictions. We employed a combination of the Medical Subject Heading (MeSH) terms, along with different spellings and endings, to identify relevant articles pertaining to "extracorporeal," "extracorporeal life support," "ECPR," "ELS," "emergency department," "emergency room," "cardiac arrest," "heart arrest," and "cardiopulmonary resuscitation". Supplementary files provide a detailed account of our search strategy. Additionally, we searched websites, organizations, relevant reviews, and references to identify additional eligible studies. Furthermore, we searched for unpublished trials registered at ClinicalTrials.gov using similar search terms. We also searched for citations from relevant articles. The search results from these databases were extracted, and duplicate studies were removed. The remaining studies were added to the Rayyan QCRI website.

## Inclusion criteria and outcome of interest

The inclusion criteria were as follows:

1. Any study included adults over 18 for whom CPR was performed at the ED.

2. At least one arm includes patients who underwent ECPR or ECLS initiated at the ED.

3. At least one arm compared with conventional or standard CPR.

4. Reporting of outcome or endpoint on the favorable neurological outcome at hospital discharge or survival to hospital discharge.

   Favorable neurological outcome was defined by a Cerebral Performance Category (CPC) of 1–2 or a modified Rankin Scale (mRS) of 0–2. We excluded pre-clinical studies, review articles, and studies without a control group (i.e., case reports, case series, etc.). Two authors (W.W. and S.K.) independently scanned and reviewed study titles and abstracts to identify potentially eligible studies. Full-text articles of the retrieved studies were extracted and independently assessed by two authors against the pre-specified criteria. Any discrepancies were discussed with another author (W.T.) and resolved through consensus.
   The primary outcome was a favorable neurological outcome at hospital discharge. We selected it as a primary outcome since the current Utstein-style guidelines recommended it for uniform reporting of cardiac arrest [25]. The secondary outcome included survival to hospital discharge/30-day survival [26].

## Data extraction and assessment of the study risk of bias

We extracted data from the included articles using a structured data extraction form, which included information such as the first author, publication year, study location, setting, enrollment period, study type, participant characteristics, criteria for intervention allocation, and outcomes of interest. In cases where specific variables of interest were missing, we attempted to contact the corresponding authors for additional details. All extracted data were entered into a data spreadsheet (Microsoft Excel). Two authors (W.W. and S.K.) independently assessed the risk of bias in the included studies, and any disagreements were resolved by discussion between authors. Quality assessment of the observational trials will be evaluated using

the Good Research for Comparative Effectiveness (GRACE) [27], and quality assessment for randomized controlled trials (RCT) will be evaluated using the Cochrane Risk of Bias 2 (RoB-2) tool [28].

## Statistical analysis

We collected the relevant information and organized it in a 2x2 contingency table in the data spreadsheet. We calculated the pooled odds ratios (ORs) and corresponding 95% confidence intervals (CIs) of the outcomes between groups using the inverse variance heterogeneity (IVhet) method. The IVhet model is known for its accuracy in estimating pooled odds ratios, especially in meta-analyses characterized by high heterogeneity [29–31]. Cochrane Q and $I^2$ statistics were used to examine the presence of heterogeneity between studies. Statistical heterogeneity among included trials was estimated using the percentage of the total variability across the studies ($I^2$ statistics). Low, moderate, and high heterogeneity were defined as values less than 25%, 25–50%, and more than 50%, respectively [32]. Sensitivity analyses were performed to investigate the benefits of ED-initiated ECPR by exclusively considering randomized controlled trials (RCTs) or studies employing propensity score (PPS) analysis.

Visual examination of Doi plots and the Luis Furuya-Kanamori (LFK) index was used to examine publication bias that may result from small-study effects [33]. MetaXL version 5.3 was used for statistical analysis (EpiGear International Pty Ltd, Australia). All tests were two-tailed, with a p-value of less than 0.05 considered statistically significant.

## Results

### Characteristics of included study and study risk of bias

Our search initially yielded a total of 1,541 articles. After removing duplicates and abstract screening, eight studies met the inclusion criteria for the meta-analysis [13–16, 34–37] (**Fig 1**). The included studies recruited participants between 2000 and 2020, totaling 51,173 individuals. The average age of participants ranged from 46 to 71. Notably, the ECPR group exhibited a higher prevalence of initial shockable rhythms than the conventional CPR group in almost all studies. Seven studies were conducted in Asia [13–16, 34, 35, 37], while one study was conducted in Europe [36]. All were observational studies [13–16, 34–37]. Four studies applied PPS matching to adjust for potential confounders. Three studies explicitly stated that emergency physicians initiated and performed ECPR [35–37]. In terms of the risk of bias, the included observational studies demonstrated a range from low to moderate risk, with two studies meeting all criteria and six studies exhibiting a moderate risk of bias. The characteristics of included studies and the risk of bias are included in **Tables 1 and 2**, respectively.

### Outcomes

Among the included studies, seven reported favorable neurological outcomes. The overall neurological outcome in the ECPR group did not significantly differ from that of the conventional CPR group (pooled OR 1.43, 95% CI 0.30–6.70, $I^2$ = 96%). However, when analyzing only PPS data, ECPR demonstrated a modest benefit (OR 1.89, 95% CI 1.26–2.83, $I^2$ = 21%). This finding was observed for studies involving emergency physician-initiated ECPR and those without (**Fig 2**).

Three studies provided data on survival to hospital discharge. ECPR patients had a higher chance of survival to hospital discharge than CCPR patients in all studies and PPS-selected data (OR 3.34, 95% CI 2.23–5.01, $I^2$ = 17% and OR 3.37, 95% CI 1.52–7.49, $I^2$ = 57%, respectively). These results were consistent across studies investigating emergency physician-initiated ECPR and those that did not (**Fig 3**).

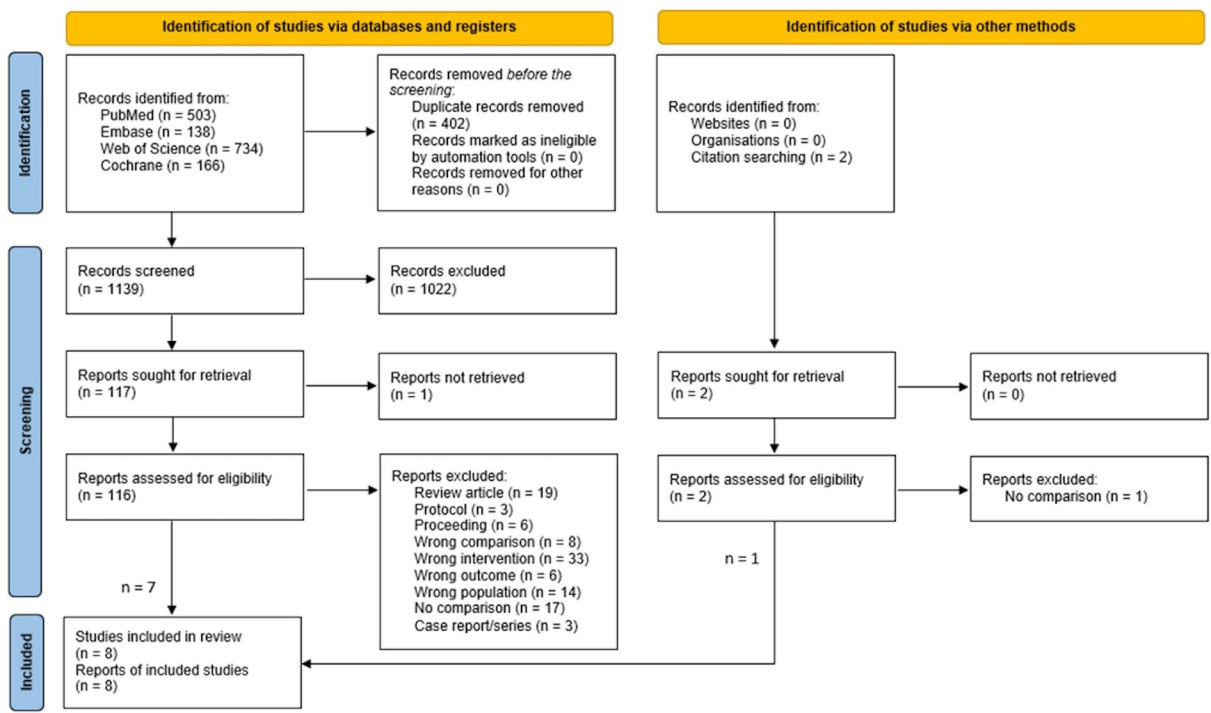

**Fig 1. PRISMA study diagram.**

## Publication bias

Doi plot analyses indicated that less precise studies carry higher increases in favorable neurological outcomes after CPR in OHCA patients (**Fig 4A**). Furthermore, major asymmetry was identified in the survival to hospital outcomes (**Fig 4B**).

## Discussion

Our systematic review and meta-analysis provide insights into the effectiveness of ECPR initiated in the ED compared to conventional CPR. While the findings regarding favorable neurological outcomes were inconclusive, there is evidence suggesting that ECPR may improve survival to hospital discharge in OHCA patients. It is important to note the presence of publication bias, which may influence the interpretation of the results.

ECMO was originally introduced to support patients with severe respiratory failure and adult respiratory distress syndrome in intensive care settings. Subsequently, it was widely applied in various life-threatening conditions, including ongoing CPR. However, due to the complex resource requirements, ECMO programs are typically limited to academic centers. Emergency physicians in many countries may be the only ones instantly available for emergent ECMO/ECPR cases. Previous evidence revealed that emergency physicians could successfully initiate ECPR in the ED with a comparable survival and neurological outcome compared to the routine ECPR conducted in the intensive care unit or cardiac catheterization laboratory [8, 10, 11, 23]. Recent evidence suggests that the faster the ECPR application, the better the resuscitation outcomes [10]. ECPR has been increasing use for patients with refractory OHCA in the past decade due to potentially improved mortality and neurological outcomes compared to conventional CPR [12, 13, 15]. Nonetheless, the efficacy and effectiveness of ECPR remain subjects of debate, with conflicting findings reported in previous research [10, 12–15].

**Table 1. Characteristics of included studies.**

| First author, year | Recruitment period, country | Study design | Participants (ECPR/CCPR) | Propensity score matching performed, participants (ECPR/CCPR) | Age, years† | | Male, n (%) | | Initial shockable rhythm, n (%) | | Witnessed arrest, n (%) | | Bystander CPR, n (%) | | Outcomes of interest |
|---|---|---|---|---|---|---|---|---|---|---|---|---|---|---|---|
| | | | | | ECPR | CCPR | ECPR | CCPR | ECPR | CCPR | ECPR | CCPR | ECPR | CCPR | |
| Choi 2016 [15] | 2009–2013, Korea | Multi-center, retrospective cohort | 320/36277 | Yes (320/320) | 56 (45–68) | 67 (54–77) | 258 (81) | 24143 (67) | 93 (29) | 3088 (9) | 228 (71) | 19630 (54) | 95 (30) | 3135 (9) | • Neurologically favorable survival at hospital discharge (CPC score of 1–2)  • Survival to hospital discharge |
| Jeong 2022 [34] | 2015–2020, Korea | Multi-center, retrospective cohort | 272/11734 | Yes (271/271) | 58 (48–67) | 71 (58–80) | 212 (78) | 7628 (65) | 163 (60) | 2341 (20) | 205 (75) | 6896 (59) | 149 (55) | 5928 (51) | • 30-day survival with favorable neurological status (CPC score of 1–2) |
| Kim 2014 [35] | 2006–2013, Korea | Single-center, prospective cohort | 55/444 | Yes (52/52) | 53 (41–68) | 69 (56–77) | 41 (75) | 285 (64) | 31 (56) | 85 (19) | 43 (78) | 328 (74) | 23 (42) | 151 (34) | • Good neurological outcome (CPC score of 1–2)  • 24-hour survival  • Survival to discharge  • Survival rate at 3 months post-arrest |
| Maekawa 2013 [37] | 2000–2004, Japan | Single-center, prospective cohort | 53/109 | Yes (24/24) | 54 (47–60) | 71 (59–80) | 44 (83) | 79 (73) | 32 (60) | 24 (22) | N/A | N/A | 29 (55) | 42 (39) | • Favorable neurological status at 3 months after cardiac arrest (CPC score of 1–2) |
| Sakamoto 2014 [16] | 2008–2012, Japan | Multi-center, prospective cohort | 260/194 | No | 56.3 ± N/A | 58.1 ± N/A | 235 (90) | 172 (89) | N/A | N/A | 186 (72) | 151 (78) | 127 (49) | 90 (46) | • Favorable neurological status at 1 month and 6 months after cardiac arrest (CPC score of 1–2) |
| Schober 2017 [36] | 2002–2012, Austria | Single-center, retrospective cohort | 7/232 | No | 46 (31–59) | 60 (50–70) | 5 (71) | 173 (75) | 4 (57) | 134 (58) | 6 (86) | 204 (88) | 2 (28) | 71 (31) | • 180 days survival in good neurological condition (CPC score of 1–2) |
| Shin 2020 [13] | 2011–2019, Korea | Single-center, before and after study | 30/40 | No | 60 (55–67) | 61 (51–70) | 25 (83) | 30 (75) | 18 (60) | 19 (48) | 18 (60) | 35 (88) | 19 (63) | 32 (80) | • Good neurological outcome at 1- and 6-months post-arrest (CPC score of 1–2) |
| Yamada 2021 [14] | 2014–2017, Japan | Multi-center, retrospective cohort | 268/878 | No | 56 (46–66) | 65 (52–73) | 231 (86) | 687 (78) | 194 (72) | 498 (57) | N/A | N/A | 135 (50) | 449 (51) | • Favorable neurological outcome at 1 month after arrest (CPC score of 1–2) |

Abbreviations: CCPR, conventional cardiopulmonary resuscitation; CPR, cardiopulmonary resuscitation; ECPR, extracorporeal cardiopulmonary resuscitation; CPC, Cerebral Performance Category; N/A, not applicable.

† Age was presented as median (interquartile range) or mean ± standard deviation.

**Table 2. Risk of bias assessment of included studies using the Good Research for Comparative Effectiveness (GRACE) checklist and Risk of Bias assessment 2 (RoB-2).**

| Observational studies | | | | | | | | | | | |
|---|---|---|---|---|---|---|---|---|---|---|---|
| Study | Adequate treatment | Adequate outcome | Objective outcomes | Valid outcomes | Similar outcomes | Covariates recorded | New initiators | Concurrent comparators | Covariates accounted for | Immortal time bias | Sensitivity analysis | Overall score (out of 11) |
| | D1 | D2 | D3 | D4 | D5 | D6 | M1 | M2 | M3 | M4 | M5 | |
| Choi 2016 | - | + | + | + | + | + | + | + | + | - | + | 9 |
| Jeong 2022 | + | + | + | + | + | + | + | + | + | - | + | 10 |
| Kim 2014 | + | + | + | + | + | + | + | + | + | + | - | 10 |
| Maekawa 2013 | + | + | + | + | + | + | + | + | + | + | + | 11 |
| Sakamoto 2014 | + | + | + | + | + | - | + | + | + | - | + | 9 |
| Schober 2017 | + | + | + | + | + | + | + | + | + | + | + | 11 |
| Shin 2020 | - | + | + | + | + | - | + | - | + | - | + | 7 |
| Yamada 2021 | - | + | + | + | + | - | + | + | + | + | + | 9 |

Despite the observed increase in favorable neurological outcomes in PPS data, the survival rate to hospital discharge for OHCA patients who underwent ECPR remains relatively low at 20.1%. These results highlight the need for further improvement, considering the expected survival rate of around 40–50% [38, 39]. The high heterogeneity observed in the forest plots suggests the presence of confounding factors. To mitigate these biases, we employed the IVhet statistical model, which gave more precise information than a conventional random-effects model [29]. However, some selection biases may persist due to the wide range of recruitment periods spanning over 20 years, during which advancements in resuscitation technologies and updated CPR guidelines were introduced. Furthermore, publication bias was identified through Doi plots and the corresponding LFK index, emphasizing the need for cautious interpretation of the pooled effect size of our findings.

This review provides the results in contrast with the previous meta-analysis [12]. Ahn et al., published in 2016, included only three OHCA studies [15, 16, 37] which found similar survival and neurological outcomes between ECPR and conventional CPR. The discrepancy can be attributed to the evolving inclusion criteria for ECPR, which have become more refined over time. Specific patient selection criteria, such as younger age, witnessed cardiac arrest with bystander CPR, initial shockable rhythm, and presumed cardiac etiology of arrest, have been proposed and likely contributed to the positive outcomes observed in subsequent studies [13, 14, 36, 40]. Moreover, patients with refractory shockable cardiac arrest have been linked with higher chances of having coronary artery disease [41, 42]. As a result, early ECPR for these patients will promote a temporary return of perfusion, mitigate the magnitude of cardiac ischemia, and provide protection from deteriorating myocardial functions to support additional resuscitation efforts until definitive interventions are delivered [43]. Therefore, emergency physician-initiated ECPR, which reduced collapse-to-ECPR time, may improve outcomes; however, optimal results would be expected when ECPR was combined with reperfusion therapy.

## Limitation

Several limitations need to be mentioned in this study. First, all included in our meta-analysis were observational studies, which may introduce selection bias, confounding, and lack of

**Primary outcome:** Favorable neurological outcome at hospital discharge

**(A) All studies**

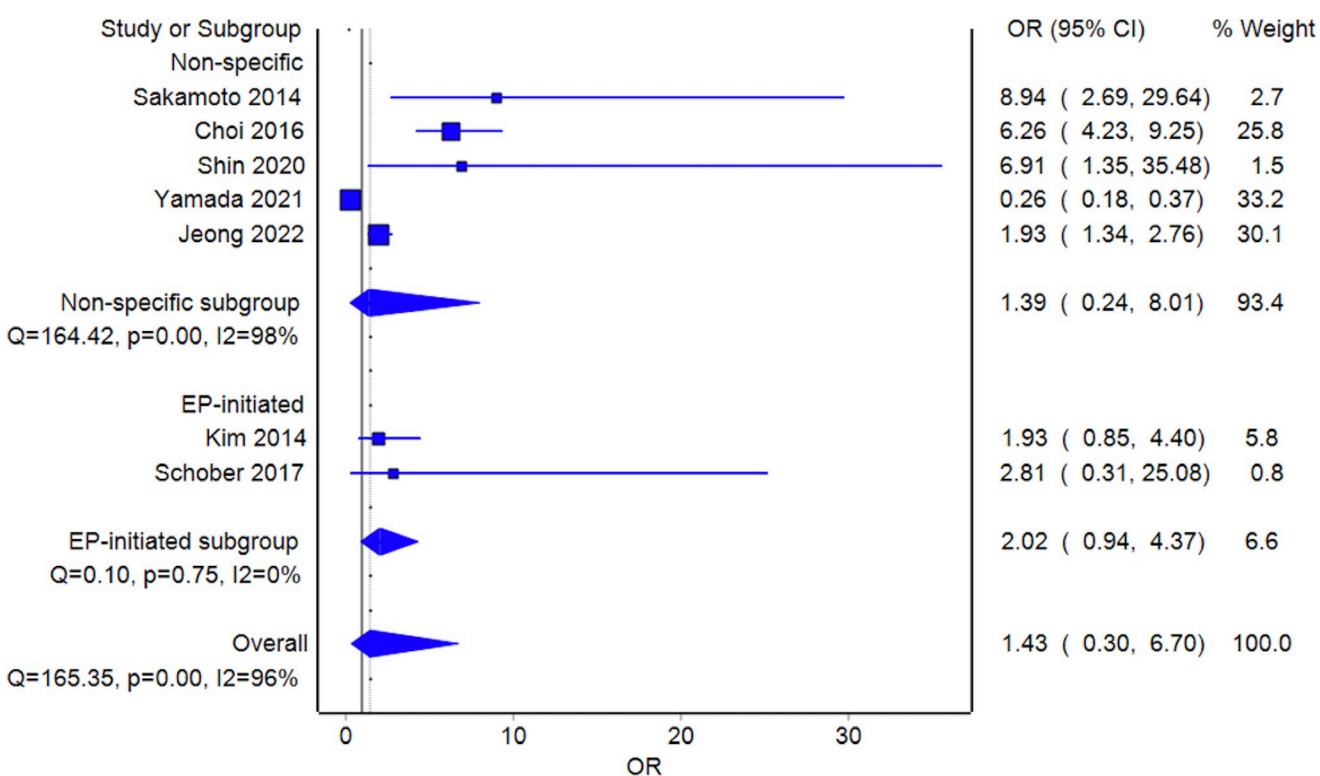

**(B) PPS data**

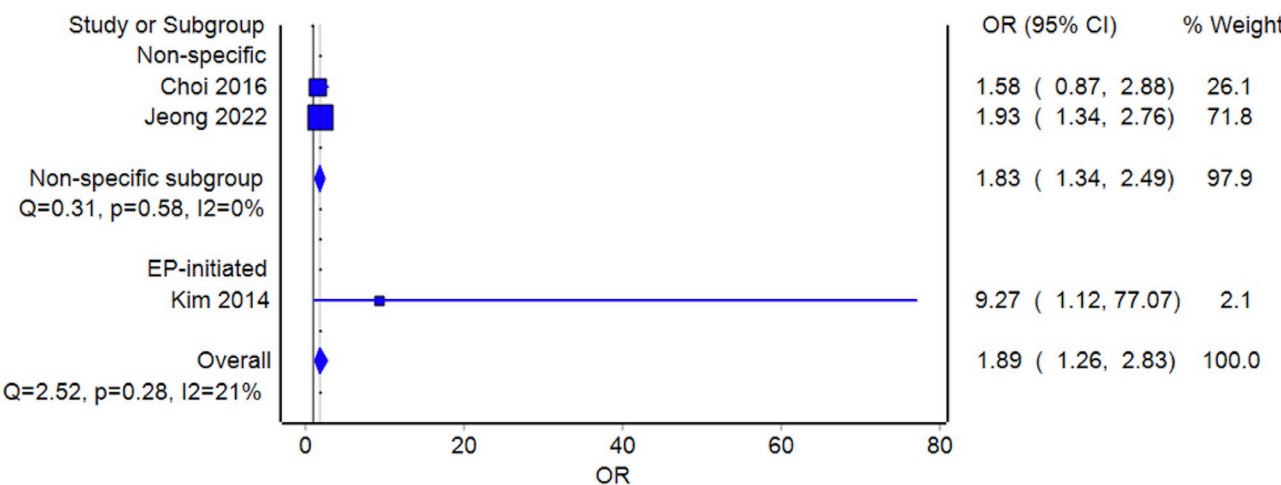

**Fig 2. Forest plots comparing favorable neurological outcomes at hospital discharge between extracorporeal cardiopulmonary resuscitation initiated at ED and conventional cardiopulmonary resuscitation.** (A) included all observational studies, and (B) selected only propensity score matching cohorts. Abbreviations: EP, emergency physician; PPS, propensity score matching; RCT, randomized controlled trial; OR, odds ratio.

**Secondary outcome:** Survival to hospital discharge

**(A) All studies**

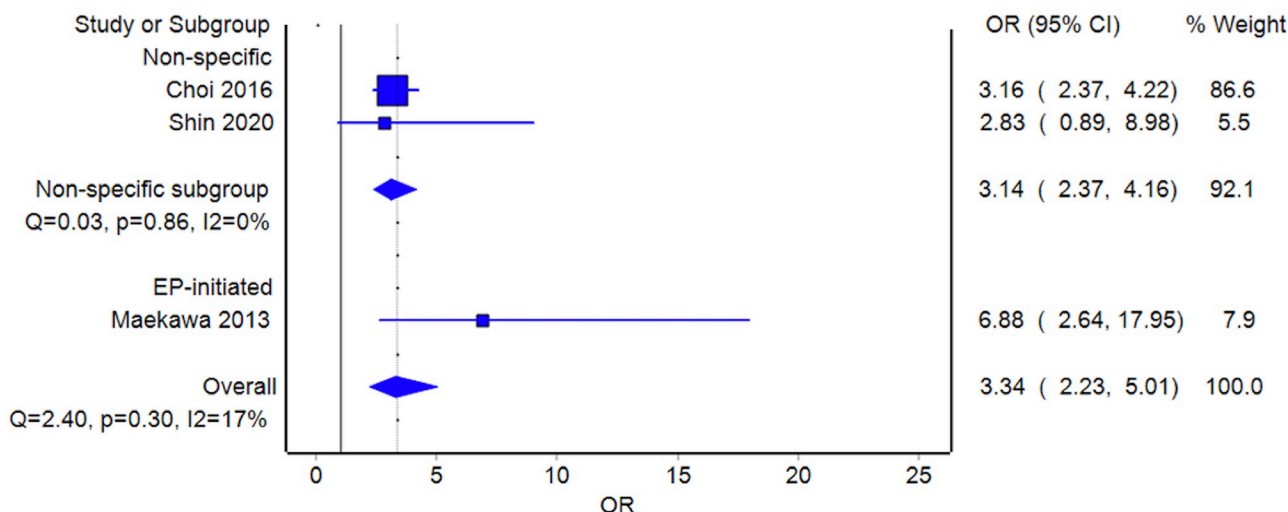

**(B) PPS data**

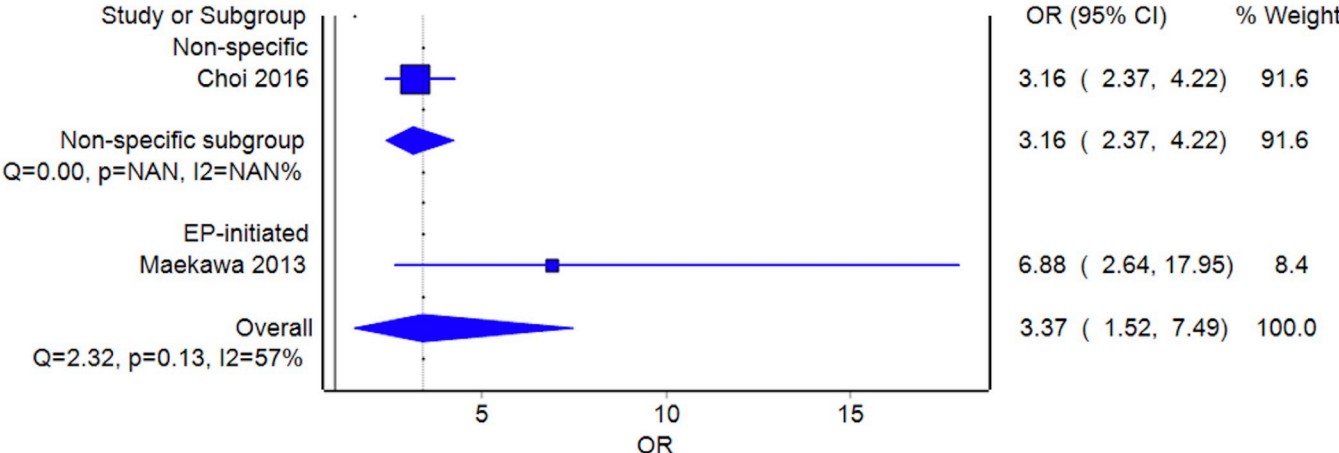

**Fig 3. Forest plots survival to hospital discharge between extracorporeal cardiopulmonary resuscitation initiated at ED and conventional cardiopulmonary resuscitation.** (A) included all observational studies, and (B) selected only propensity score matching cohorts. Abbreviations: EP, emergency physician; PPS, propensity score matching; RCT, randomized controlled trial; OR, odds ratio.

control groups. Furthermore, most of the studies were conducted at single centers, limiting the generalizability of the findings to other settings. The observed heterogeneity among the included studies also poses challenges in drawing definitive conclusions. Future studies should prioritize conducting randomized controlled trials with larger sample sizes and aim to achieve homogeneity in selected patient populations and coexisting treatments. Second, included studies did not provide information on logistics and human resources, which are known to have a strong impact on the survival rate of CPR. It is commonly known that important CPR-related parameters include the time for an ambulance to arrive and the time to administer the first shock from an automated external defibrillator. Still, only some studies included in the review

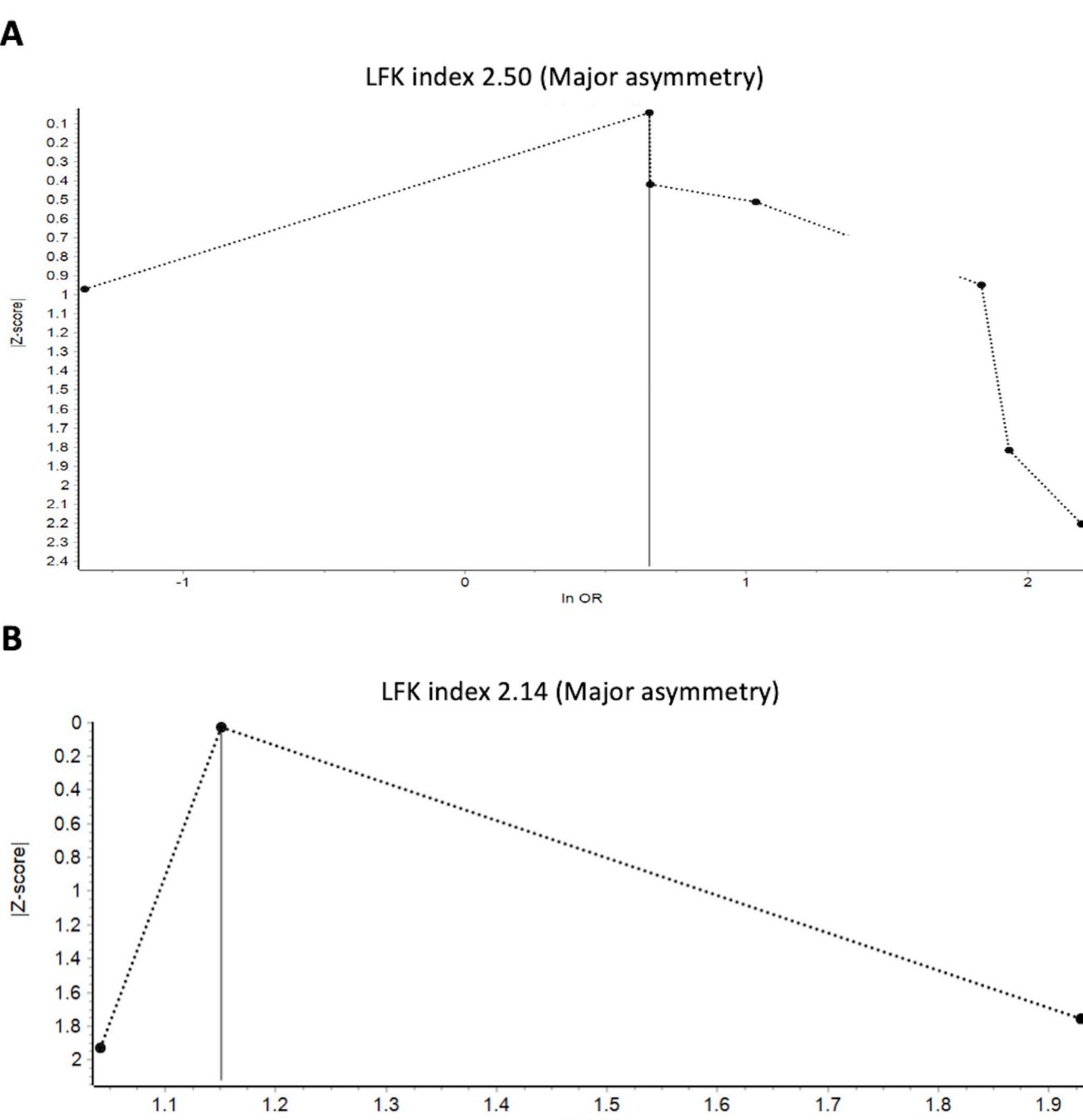

**Fig 4.** Doi plots for studies reporting favorable neurological outcomes at hospital discharge (A) and survival to hospital discharge (B). Abbreviations: LFK = Luis Furuya-Kanamari; OR = odds ratio.

report this information. Additionally, the review did not provide insight into other important factors, such as the willingness of the general public to perform CPR or the training of hospital teams. Third, complications arising from ECPR (i.e., significant bleeding, coagulopathy, metabolic derangements, etc.), which can influence neurologically intact survival and survival to discharge, were only reported in one study. Finally, the geographical restriction of the included studies to South Korea and Japan raises concerns regarding the generalizability of the findings

to other regions and populations. Race and ethnicity issues and low multi-center studies could contribute to restrict the representativeness of the results. Future studies should strive to include larger and more diverse samples to ensure more robust conclusions.

## Conclusion

In conclusion, our systematic review and meta-analysis demonstrated conflicting results regarding the effectiveness of ECPR initiated in the ED. While there is limited evidence supporting improved neurological outcomes, ECPR may confer benefits in terms of survival to hospital discharge for OHCA patients. Future studies should focus on randomized controlled trials with larger sample sizes and aim to achieve homogeneity in patient populations.

## Supporting information

**S1 Checklist. RISMA 2020 checklist.**
(DOCX)

**S1 Table. Electronic search terms.**
(DOCX)

## Author Contributions

**Conceptualization:** Wachira Wongtanasarasin, Sarunsorn Krintratun, Witina Techasatian, Daniel K. Nishijima.

**Data curation:** Wachira Wongtanasarasin.

**Formal analysis:** Wachira Wongtanasarasin, Sarunsorn Krintratun.

**Investigation:** Wachira Wongtanasarasin, Sarunsorn Krintratun.

**Methodology:** Sarunsorn Krintratun, Daniel K. Nishijima.

**Supervision:** Wachira Wongtanasarasin, Daniel K. Nishijima.

**Visualization:** Daniel K. Nishijima.

**Writing – original draft:** Wachira Wongtanasarasin.

**Writing – review & editing:** Wachira Wongtanasarasin, Sarunsorn Krintratun, Witina Techasatian, Daniel K. Nishijima.

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
