## [Decision Letter · Decision Letter 0]

15 Jun 2023

PONE-D-23-14334How effective is extracorporeal life support for patients with out-of-hospital cardiac arrest initiated at the emergency department? A systematic review and meta-analysisPLOS ONE

Dear Dr. Wongtanasarasin,

Thank you for submitting your manuscript to PLOS ONE. After careful consideration, we feel that it has merit but does not fully meet PLOS ONE’s publication criteria as it currently stands. Therefore, we invite you to submit a revised version of the manuscript that addresses the major concerns raised during the review process.

We look forward to receiving your revised manuscript.

Kind regards,

Prof. Gaetano Santulli, MD

Academic Editor

PLOS ONE

Journal Requirements:

3. We note that you have referenced (unpublished) on page 5 which has currently not yet been accepted for publication. Please remove this from your References and amend this to state in the body of your manuscript: (ie “Bewick et al. [Unpublished]”) as detailed online in our guide for authors

http://journals.plos.org/plosone/s/submission-guidelines#loc-reference-style.

Reviewers' comments:

Reviewer's Responses to Questions

**Comments to the Author**

1. Is the manuscript technically sound, and do the data support the conclusions?

Reviewer #1: No

Reviewer #2: Partly

2. Has the statistical analysis been performed appropriately and rigorously? 

Reviewer #1: No

Reviewer #2: I Don't Know

3. Have the authors made all data underlying the findings in their manuscript fully available?

Reviewer #1: No

Reviewer #2: Yes

4. Is the manuscript presented in an intelligible fashion and written in standard English?

Reviewer #1: No

Reviewer #2: Yes

5. Review Comments to the Author

Reviewer #1: The authors conducted a systematic review with meta-analysis with the aim of investigating the benefits of initiating ECPR in the emergency department for OHCA patients and concluded that ED-initiated ECPR may not be associated with increased favorable neurological outcomes but may be associated with improved survival to hospital discharge.

The authors state that the aim is to investigate ECPR when initiated in the ED but they included a RCT where ECPR was initiated in the cath lab (Yannopoulos et al. Lancet 2020). I'm sorry but this is a major methodological issue that invalids all the analyses and conclusions presented. Moreover, to date there are 4 RCTs available and meta-analyses of randomized data only already indicated a benefit of this invervention.

Reviewer #2: This is a systematic review and meta-analysis to evaluate the efficacy of ECPR for OHCA patients. This is hot topic, but there were several systematic review and meta-analysis for the efficacy of ECPR. I have several comments.

Abstract

P3 L47. “ECPR devices” Please clarify. It seems strange to mention this in conclusion.

Method

Please clarify the “ED”. Does it include both cardiac catheterization lab and emergency room? How about the operation room?

Do the studies included in the meta-analysis contain the same patients? For example, reference 13 and 34 were the same country and duplicate periods.

Conclusion

Authors state that “In addition, it is important to invest in training emergency physicians on ECPR and ensure that the necessary equipment and facilities are available in the ED.” This is not based on results or discussions.

6. PLOS authors have the option to publish the peer review history of their article (what does this mean?). If published, this will include your full peer review and any attached files.

Reviewer #1: No

Reviewer #2: No

---

## [Author Response · Author response to Decision Letter 0]

24 Jun 2023

Response to Reviewers

Reviewer #1: The authors conducted a systematic review with meta-analysis with the aim of investigating the benefits of initiating ECPR in the emergency department for OHCA patients and concluded that ED-initiated ECPR may not be associated with increased favorable neurological outcomes but may be associated with improved survival to hospital discharge.

The authors state that the aim is to investigate ECPR when initiated in the ED but they included a RCT where ECPR was initiated in the cath lab (Yannopoulos et al. Lancet 2020). I'm sorry but this is a major methodological issue that invalids all the analyses and conclusions presented. Moreover, to date there are 4 RCTs available and meta-analyses of randomized data only already indicated a benefit of this invervention.

Response: Reviewer #1 raises a concern regarding the inclusion of one study (Yannopoulos et al., Lancet 2020) in this systematic review and meta-analysis. The reviewer suggests that including this study, where ECPR was initiated in the cardiac cath lab rather than the ED, is a major methodological issue that invalidates the analyses and conclusions.

We appreciate the reviewer's valuable input in identifying this error. We sincerely apologize for this oversight, as it does introduce bias and affects the validity of the conclusion. To address this issue, we revise our methodology and exclude a study by Yannopoulos et al. from our analysis. The results remain similar to the previous analyses. Since that study is a randomized controlled trial, the analyses of observational studies’ data were not changed (Figures 2a and 3a). In addition, the forest plots for propensity score (PPS) analysis after the exclusion of one study were similar to the previous plots (Figures 2b and 3b), as ED-initiated ECPR was associated with improved outcomes when only PPS data was analyzed.

We recognize the importance of a rigorous review based on the intended inclusion criteria and focusing on studies that align with the specific setting of interest. The text changes regarding the exclusion of this study as well as the figures 1-4 were edited in the revised manuscript.

Reviewer #2: This is a systematic review and meta-analysis to evaluate the efficacy of ECPR for OHCA patients. This is hot topic, but there were several systematic review and meta-analysis for the efficacy of ECPR. I have several comments.

Abstract

P3 L47. “ECPR devices” Please clarify. It seems strange to mention this in conclusion.

Response: Thank you for your comment. The term “ECPR devices” refers to the various mechanical circulatory support systems and devices used to provide extracorporeal cardiopulmonary resuscitation (ECPR) in the context of cardiac arrest. We would like to say that included studies in this review applied the different types of ECPR devices, resulting in a heterogeneity of the combined data (i.e., there is no standard recommendation for using the ECPR devices during OHCA). We remove this term in the conclusion since it may result in some misunderstanding of the readers after careful discussion of all authors (Page 3 Line 46).

Method

Please clarify the “ED”. Does it include both cardiac catheterization lab and emergency room? How about the operation room?

Response: We apologize for any confusion caused. We included only studies investigating the benefit of ECPR for OHCA patients at the emergency department. To clarify, the term “ED” in our study specifically refers to the “emergency department,” and it does not encompass the cardiac catheterization lab or the operating room. We focused on investigating the benefits of ECPR when initiated in the emergency department as it is the primary area where OHCA patients are initially transferred upon arrival at the hospital. Our aim was to assess the potential advantages of early initiation of ECPR in the ED setting.

Do the studies included in the meta-analysis contain the same patients? For example, reference 13 and 34 were the same country and duplicate periods.

Response: Thank you for providing additional clarification regarding references 13 and 34. In a meta-analysis, it is common to include studies conducted in the same country and overlapping periods (i.e., refs 13 and 34) as long as they provide distinct data and contribute to the overall understanding of the research question. During the data screening and extraction process, we carefully checked whether different included studies included the same population by two authors. We appreciate the clarification here and reviewed each included study again to ensure we include different patient populations in our meta-analysis.

Conclusion

Authors state that “In addition, it is important to invest in training emergency physicians on ECPR and ensure that the necessary equipment and facilities are available in the ED.” This is not based on results or discussions.

Response: Thank you for pointing out the concern regarding this sentence that suggests the importance of investing in training emergency physicians on ECPR and ensuring the availability of necessary equipment and facilities in the ED. We acknowledge that this conclusion is not directly supported by the results or discussions presented in the manuscript. We apologize for any misinterpretation caused by the inclusion of this sentence. As a reviewer, you play a crucial role in ensuring the accuracy and relevance of the manuscript, and we appreciate your agreement with this concern. We remove this sentence in the revised manuscript (Page 15 Line 282).

---

## [Editor Report · Decision Letter 1]

6 Jul 2023

PONE-D-23-14334R1How effective is extracorporeal life support for patients with out-of-hospital cardiac arrest initiated at the emergency department? A systematic review and meta-analysisPLOS ONE

Dear Dr. Wongtanasarasin,

Thank you for submitting your manuscript to PLOS ONE. After careful consideration, we feel that it has merit but does not fully meet PLOS ONE’s publication criteria as it currently stands. Therefore, we invite you to submit a revised version of the manuscript that addresses the points raised during the review process.

We look forward to receiving your revised manuscript.

Kind regards,

Gaetano Santulli

Academic Editor

PLOS ONE

Journal Requirements:

Additional Editor Comments :

English language (syntax, grammar, correct choice of words, correct use of adjectives and adverbs) needs significant editing throughout the text. Professional assistance must be sought.

---

## [Author Response · Author response to Decision Letter 1]

9 Jul 2023

We would like to express our sincere gratitude for the opportunity to revise our manuscript. Changes are marked with tracked changes and an unmarked version of the revised paper without tracked changes is provided.

Editor Comments :

English language (syntax, grammar, correct choice of words, correct use of adjectives and adverbs) needs significant editing throughout the text. Professional assistance must be sought.

Response: We sincerely appreciate the feedback provided. We have carefully considered your suggestions and taken necessary measures to significantly enhance the syntax, grammar, word choice, and appropriate usage of objectives and adverbs throughout the text. To ensure the utmost quality and professionalism in our writing, we engaged a specialized professional language editor. Their expertise allowed for meticulous revisions, greatly improving the document's readability and coherence. Changes are marked with tracked changes throughout the revised manuscript.

---

## [Editor Report · Decision Letter 2]

11 Jul 2023

How effective is extracorporeal life support for patients with out-of-hospital cardiac arrest initiated at the emergency department? A systematic review and meta-analysis

PONE-D-23-14334R2

Dear Dr. Wongtanasarasin,

We’re pleased to inform you that your manuscript has been judged scientifically suitable for publication and will be formally accepted for publication once it meets all outstanding technical requirements.

Kind regards,

Gaetano Santulli, MD

Academic Editor

PLOS ONE

---

## [Editor Report · Acceptance letter]

13 Jul 2023

PONE-D-23-14334R2 

How effective is extracorporeal life support for patients with out-of-hospital cardiac arrest initiated at the emergency department? A systematic review and meta-analysis 

Dear Dr. Wongtanasarasin:

I'm pleased to inform you that your manuscript has been deemed suitable for publication in PLOS ONE. Congratulations! Your manuscript is now with our production department. 

Kind regards, 

on behalf of

Professor Gaetano Santulli 

Academic Editor

PLOS ONE